# Structural ordering of the *Plasmodium berghei* circumsporozoite protein repeats by inhibitory antibody 3D11

Iga Kucharska[1†], Elaine Thai[1,2†], Ananya Srivastava[1,2], John L Rubinstein[1,2,3], Régis Pomès[1,2], Jean-Philippe Julien[1,2,4*]

[1]Program in Molecular Medicine, The Hospital for Sick Children Research Institute, Toronto, Canada; [2]Department of Biochemistry, University of Toronto, Toronto, Canada; [3]Department of Medical Biophysics, University of Toronto, Toronto, Canada; [4]Department of Immunology, University of Toronto, Toronto, Canada

**\*For correspondence:**
jean-philippe.julien@sickkids.ca

[†]These authors contributed equally to this work

**Competing interests:** The authors declare that no competing interests exist.

**Abstract** Plasmodium sporozoites express circumsporozoite protein (CSP) on their surface, an essential protein that contains central repeating motifs. Antibodies targeting this region can neutralize infection, and the partial efficacy of RTS,S/AS01 – the leading malaria vaccine against *P. falciparum* (Pf) – has been associated with the humoral response against the repeats. Although structural details of antibody recognition of PfCSP have recently emerged, the molecular basis of antibody-mediated inhibition of other Plasmodium species via CSP binding remains unclear. Here, we analyze the structure and molecular interactions of potent monoclonal antibody (mAb) 3D11 binding to *P. berghei* CSP (PbCSP) using molecular dynamics simulations, X-ray crystallography, and cryoEM. We reveal that mAb 3D11 can accommodate all subtle variances of the PbCSP repeating motifs, and, upon binding, induces structural ordering of PbCSP through homotypic interactions. Together, our findings uncover common mechanisms of antibody evolution in mammals against the CSP repeats of Plasmodium sporozoites.

## Introduction

Despite extensive biomedical and public health measures, malaria persists as a major global health concern, with an estimated 405,000 deaths and 228 million cases annually (*WHO, 2019*). Moreover, resistant strains have been detected against all currently available antimalarial drugs, including sulfadoxine/pyrimethamine, mefloquine, halofantrine, quinine, and artemisinin (*Cui et al., 2015*; *Ross and Fidock, 2019*). Although ~94% of deaths are caused by *Plasmodium falciparum* (Pf) (*WHO, 2019*), other Plasmodium species that infect humans (*P. vivax, P. malariae, P. knowlesi* and *P. ovale*) also cause debilitating disease and have been associated with fatal outcomes (*Lover et al., 2018*). All Plasmodium species have a complex life cycle divided between a vertebrate host and an *Anopheles* mosquito vector (*Hall and Fauci, 2009*). During a blood meal, sporozoites are deposited into the skin of a host organism from the salivary glands of a mosquito, and subsequently migrate through the bloodstream to infect host hepatocytes (*de Koning-Ward et al., 2015*). Due to the small number of parasites transmitted and the expression of protein antigens that possess conserved functional regions (*Rosenberg et al., 1990*; *Smith et al., 2014*), the pre-erythrocytic sporozoite stage of the Plasmodium life cycle has long been considered a promising target for the development of an anti-malarial vaccine (*Nussenzweig and Nussenzweig, 1984*).

Circumsporozoite protein (CSP) is the most abundant protein on the surface of Plasmodium sporozoites, and is necessary for parasite development in mosquitoes and establishment of infection in host liver cells (*Cerami et al., 1992*; *Frevert et al., 1993*; *Ménard et al., 1997*). Flanked by N- and C-terminal domains, CSP contains an unusual central region consisting of multiple, short (4 to 8)

**eLife digest** Malaria is a significant health concern, killing about 400,000 people each year. While antimalarial drugs and insecticides have successfully reduced deaths over the last 20 years, the parasite that causes malaria is starting to gain resistance to these treatments. Vaccines offer an alternative route to preventing the disease. However, the most advanced vaccine currently available provides less than 50% protection.

Vaccines work by encouraging the body to develop proteins called antibodies, which can recognize the parasite and trigger an immune response that blocks the infection. These antibodies target a molecule on the parasite's surface called circumsporozoite protein, or CSP for short. Therefore, having a better understanding of how antibodies interact with CSP could help researchers design more effective treatments.

A lot of what is known about malaria has come from studying this disease in mice. However, it remained unclear whether antibodies produced in rodents combat the malaria-causing parasite in a similar manner to human antibodies. To answer this question, Kucharska, Thai et al. studied a mouse antibody called 3D11, which targets CSP on the surface of a parasite that causes malaria in rodents. The interaction between CSP and 3D11 was studied using three different techniques in order to better understand how the structure of CSP changes when bound by antibodies.

The experiments showed that although CSP has a highly flexible structure, it forms a more stable, spiral-like architecture when bound to multiple copies of 3D11. A similar type of assembly was previously observed in studies investigating how CSP interacts with human antibodies. Further investigation revealed that the molecular connections between 3D11 and CSP share a lot of similarities with how human antibodies recognize CSP.

These findings reveal how mammals evolved similar mechanisms for detecting and inhibiting malaria-causing parasites. This highlights the robust features antibodies need to launch an immune response against malaria, which could help develop a more effective vaccine.

amino acid (aa) repeats (*Eichinger et al., 1986*; *Dame et al., 1984*; *Plassmeyer et al., 2009*; *Zavala et al., 1983*). The sequence of the repeating motif depends on the Plasmodium species and field isolate (*Chenet et al., 2012*; *Rich et al., 2000*; *Tahar et al., 1998*). Importantly, the central region of CSP is highly immunodominant and antibodies targeting the repeats can inhibit sporozoite infectivity by preventing parasite migration (*Mishra et al., 2012*) and attachment to hepatocytes (*Potocnjak et al., 1980*; *Yoshida et al., 1980*). PfCSP is a major component of the leading malaria vaccine RTS,S/AS01, which is currently undergoing pilot implementation in Africa (*Adepoju, 2019*; *Draper et al., 2018*). Anti-PfCSP repeat antibodies have been suggested to form the predominant humoral immune response elicited by RTS,S/AS01, and correlate with vaccine efficacy (*Dobaño et al., 2019*; *McCall et al., 2018*; *Olotu et al., 2016*). However, RTS,S/AS01 offers only modest and short-lived protection (*RTS,S Clinical Trials Partnership, 2015*; *RTS,S Clinical Trials Partnership et al., 2012*; *RTS,S Clinical Trials Partnership et al., 2011*); thus, it is critical to develop a better molecular understanding of the antibody response against this Plasmodium antigen, particularly the repeat region (*Davies et al., 2015*; *Doolan, 2011*; *Illingworth et al., 2019*), to obtain valuable information needed for improved vaccine design.

Our understanding of Plasmodium biology and key host-parasite interactions has been enhanced by studies using rodent parasites, including *P. berghei* (Pb), *P. chabaudi* and *P. yoelii* (*De Niz and Heussler, 2018*). In vivo studies evaluating the inhibitory potential of mAbs are often derived from these rodent parasite models, or transgenic rodent sporozoites harboring PfCSP, as Pf fails to infect rodents. For example, mAb 3D11 was isolated from mice exposed to the bites of mosquitoes that had been infected with γ-irradiated Pb parasites (*Yoshida et al., 1980*). mAb 3D11 recognition of the PbCSP central repeat region on the surface of live sporozoites resulted in abolished Pb infectivity in vitro and in vivo (*Cochrane et al., 1976*). Electron micrographs of Pb sporozoites pre-treated with mAb 3D11 revealed the presence of amorphous, precipitated material on the parasite surface characteristic of the circumsporozoite precipitation reaction (*Yoshida et al., 1980*). This antibody continues to be widely used in model systems of sporozoite infection. For example, a recent study used mAb 3D11 in combination with transmission-blocking mAb 4B7 to show that antibody targeting of

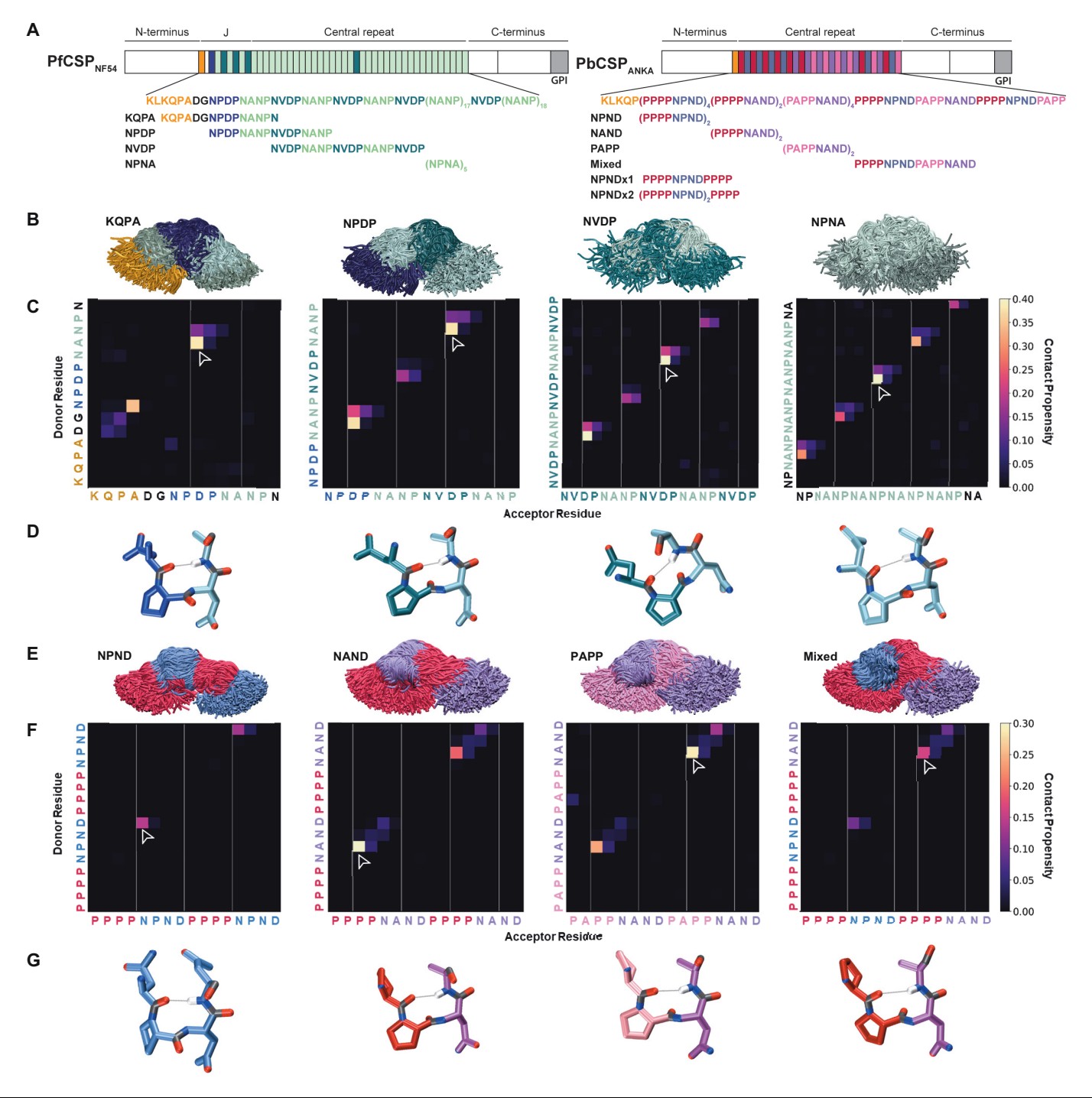

**Figure 1.** Comparison of PfCSP and PbCSP repeat sequences and structures. (**A**) Schematic representations of PfCSP strain NF54 and PbCSP strain ANKA, each comprising an N-terminal domain, central repeat region, and C-terminal domain. The junctional region (J) immediately following the N-terminal domain of PfCSP is indicated. Colored bars represent each repeat motif. The sequences of each CSP central repeat region and corresponding peptides used in the study are shown below their respective schematics. (**B-G**) Conformational ensembles of CSP peptides in solution from molecular dynamics simulations. (**B**) Superposition of the conformations of the four PfCSP-derived peptides at each nanosecond. The peptides are aligned to the conformational median structure and only the backbone is shown for clarity. (**C**) Ensemble-averaged backbone-backbone hydrogen-bonding maps for each PfCSP peptide sequence. The propensity for hydrogen bonds between the NH groups (y-axis) and CO groups (x-axis) is indicated by the color scale on the right. (**D**) Sample molecular dynamics snapshots of the highest-propensity turn for each PfCSP peptide are shown as sticks with hydrogen bonds shown as gray lines. The highest-propensity turn for each peptide is indicated by the arrowhead on the corresponding

*Figure 1 continued on next page*

*Figure 1 continued*

hydrogen-bonding map. (**E**) Superposition of the conformations of the four PbCSP-derived peptides at each nanosecond. The peptides are aligned to the conformational median structure and only the backbone is shown for clarity. (**F**) Ensemble-averaged backbone-backbone hydrogen-bonding maps for each PbCSP peptide sequence. The propensity for hydrogen bonds between the NH groups (y-axis) and CO groups (x-axis) is indicated by the color scale on the right. (**G**) Sample molecular dynamics snapshots of the highest-propensity turn for each PbCSP peptide are shown as sticks with hydrogen bonds shown as gray lines. The highest-propensity turn for each peptide is indicated by the arrowhead on the corresponding hydrogen-bonding map. The online version of this article includes the following figure supplement(s) for figure 1:

**Figure supplement 1.** Ensemble-averaged hydrogen-bonding propensities for PfCSP- and PbCSP-derived peptides.
**Figure supplement 2.** Experimental details of MD simulations.

both the pre-erythrocytic and sexual stages of a Pfs25-transgenic Pb parasite led to a synergistic reduction of parasite transmission in mice (*Sherrard-Smith et al., 2018*). However, it remains unclear whether murine mAb 3D11 recognizes the central domain of PbCSP with the same molecular principles as the most potent human anti-PfCSP repeat antibodies, for which molecular details have recently emerged (*Imkeller et al., 2018*; *Julien and Wardemann, 2019*; *Kisalu et al., 2018*; *Murugan et al., 2020*; *Oyen et al., 2018*; *Oyen et al., 2017*; *Tan et al., 2018*; *Triller et al., 2017*).

Here, we characterized the structure of the PbCSP repeats unliganded and as recognized by mAb 3D11. Our molecular studies reveal that mAb 3D11 binds across all PbCSP repeat motifs and induces structural ordering of PbCSP in a spiral-like conformation using homotypic interactions.

## Results

### Repeat motifs of PfCSP and PbCSP have similar structural propensities

The central repeats of PfCSP and PbCSP consist of recurring 4-aa motifs rich in asparagine and proline residues (*Figure 1A*). PfCSP is composed of repeating NANP motifs interspersed with intermittent NVDP repeats, and a singular NPDP motif in the junction immediately following the N-terminal domain. While the major repeat motif of PfCSP is often referred to as NANP, numerous reports have identified NPNA as the structurally relevant unit of the central region (*Kisalu et al., 2018*; *Oyen et al., 2017*; *Dyson et al., 1990*; *Ghasparian et al., 2006*). Similarly, the central domain of PbCSP contains an array of PPPP and PAPP motifs interspersed with NPND or NAND motifs (*Figure 1A*). Notably, both orthologs contain the conserved pentamer, KLKQP, known as Region I, at the C-terminal end of the N-terminal domain.

To examine and compare the structural properties of the various Pf and Pb repeat motifs in solution, we performed molecular dynamics (MD) simulations using eight different peptides ranging in length from 15 to 20 aa, with four peptides derived from Pf [KQPADGNPDPNANPN ('KQPA'); NPDPNANPNVDPNANP ('NPDP'); (NVDPNANP)$_2$NVDP ('NVDP'); and (NPNA)$_5$ ('NPNA')], and four peptides from Pb [(PPPPNPND)$_2$ ('NPND'); (PPPPNAND)$_2$ ('NAND'); (PAPPNAND)$_2$ ('PAPP'); and PPPPNPNDPAPPNANAD ('Mixed'); *Figure 1B–G*]. Each simulation was conducted in water for a total production time of 18 µs. All eight peptides were highly disordered and adopted a large ensemble of conformations with low to moderate secondary structure propensities (*Figure 1B and E*), which are best described in statistical terms. The only secondary structure observed was local, and consisted of sparse, transient hydrogen-bonded turns (*Figure 1C and F*, *Figure 1—figure supplement 1* and *Figure 1—figure supplement 2*, and *Supplementary file 1*). In particular, these interactions consisted of forward α-, β-, and γ-turns, with the β-turns being the most populated (up to 40%; *Figure 1C and F*), consistent with previous NMR studies focused on the NANP repeats (*Dyson et al., 1990*). Across all peptides, the average β-turn lifetime ranged from 2.7 ± 0.2 ns for the Pb PPNA turn to 4.4 ± 0.4 ns for DPNA turns found within PfCSP (*Supplementary file 1*).

In line with reports identifying NPNA as the main structural repeating unit of PfCSP (*Kisalu et al., 2018*; *Oyen et al., 2017*; *Dyson et al., 1990*; *Ghasparian et al., 2006*), turns were predominantly observed within these motifs, as well as DPNA, NPNV, and ADGN sequences amongst the PfCSP peptides. NPND and PPNA exhibited the greatest propensity to form β-turns of the PbCSP repeats. Importantly, each individual motif consistently exhibited the same structural tendencies, independent of their position and the overall peptide sequence in which they were contained (*Figure 1C and F*, and *Supplementary file 1*). Furthermore, using the probability rule stating that two events

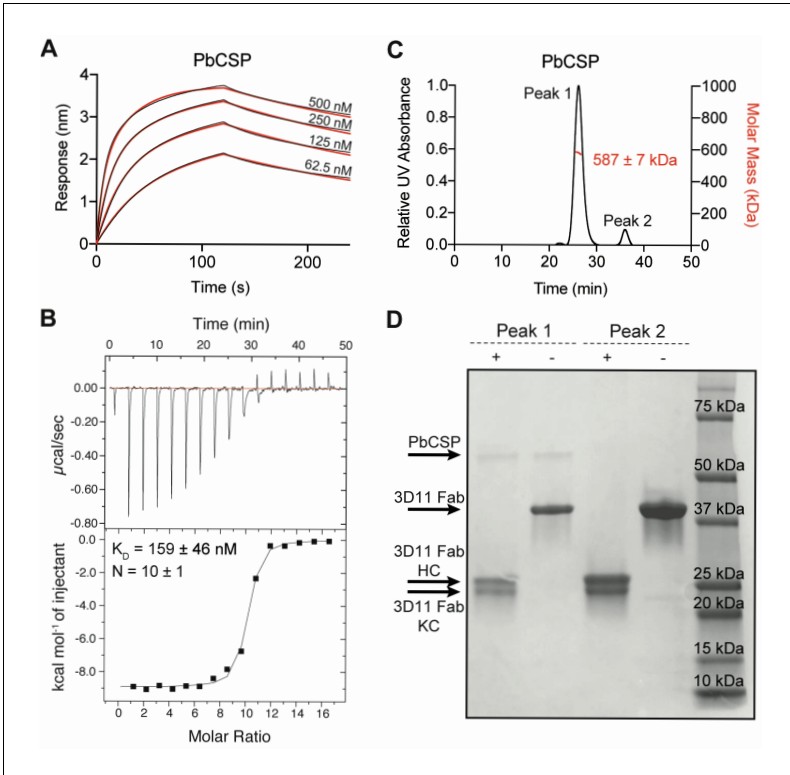

**Figure 2.** Biophysical characterization of 3D11 Fab-PbCSP binding. (A) Binding kinetics of twofold dilutions of 3D11 Fab to PbCSP. Representative sensorgrams are shown in black and 2:1 model best fits in red. Data are representative of three independent measurements. (B) Isothermal titration calorimetry (ITC) analysis of 3D11 Fab binding to PfCSP at 37°C. Above, raw data of PbCSP (0.005 mM) in the sample cell titrated with 3D11 Fab (0.4 mM). Below, plot and trendline of heat of injectant corresponding to the raw data. $K_D$ and N values resulting from three independent experiments are indicated. Standard error values are reported as standard error of the mean (SEM). (C) Results from size-exclusion chromatography coupled with multi-angle light scattering (SEC-MALS) for the 3D11 Fab-PbCSP complex. A representative measurement of the molar mass of the 3D11 Fab-PbCSP complex is shown as the red line. Mean molar mass and standard deviation are as indicated. (D) SDS-PAGE analysis of resulting Peaks 1 and 2 from SEC-MALS. Each peak was sampled in reducing and non-reducing conditions as indicated by + and -, respectively.

are independent if the equation P(A∩B)=P(A)·P(B) holds true, we show that the presence of an intra-molecular hydrogen bond in one motif does not alter the hydrogen-bonding propensities of adjacent motifs (*Figure 1—figure supplement 2B and C*). Therefore, in both PfCSP and PbCSP we conclude that there is no discernable cooperativity between the structures of different repeat motifs, and as such, in the absence of extended or nonlocal secondary structure, each of these motifs behaves as an independent unit with its own intrinsic secondary structure propensities.

To examine the influence of Asn, Asp, and Gln sidechains on the conformational ensemble of the peptides, we computed contact maps for backbone-sidechain hydrogen bonds (*Figure 1—figure supplement 1* and *Supplementary file 1*). We found that the majority of contacts are in the form of pseudo α-turns and β-turns, with backbone NH groups donating to sidechain O atoms. Notably, we discovered that these transient sidechain contacts do not have a stabilizing effect on backbone-backbone hydrogen bonds and consequently, are not correlated with the presence of these bonds (numerical example in *Figure 1—figure supplement 2B and C*).

In summary, the four Pf and four Pb peptides corresponding to CSP central repeats were all found to be highly disordered, resulting in an ensemble of conformations. The only secondary structure elements present were sparse and local hydrogen-bonded turns within each motif. Each structural motif acted independently from adjacent sequences and behaved similarly in various peptides.

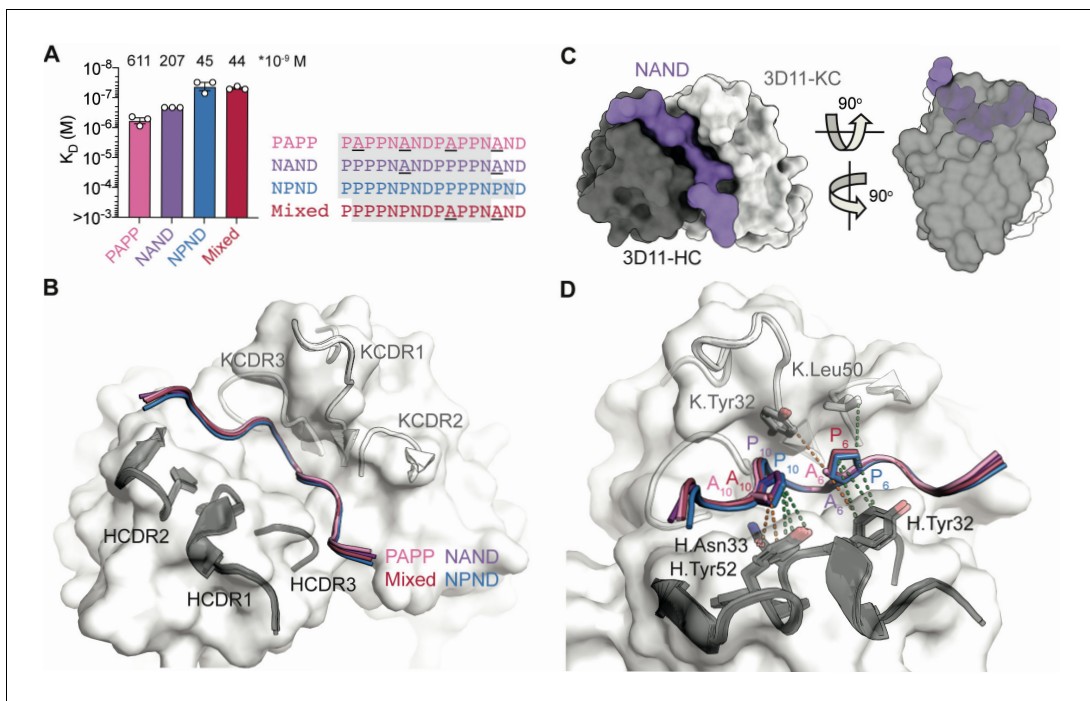

**Figure 3.** 3D11 Fab binding to PbCSP repeat peptides. (**A**) Affinities of 3D11 Fab for PAPP, NAND, NPND, and Mixed peptides as measured by ITC. Symbols represent independent measurements. Mean $K_D$ values are shown above the corresponding bar. Error bars represent SEM. Peptide sequences are as indicated to the right of the plot, with variable residues underlined and shaded residues indicating those resolved in the corresponding X-ray crystal structures. (**B**) The 3D11 Fab binds the PAPP (pink), NAND (purple), NPND (blue) and Mixed (red) peptides in nearly identical conformations. mAb 3D11 CDRs are indicated. (**C**) Overview and side view of the NAND peptide (purple) in the binding groove of the 3D11 Fab shown as surface representation (H-chain shown in black and K-chain shown in gray). (**D**) Van der Waals interactions formed by side chain atoms of both Ala and Pro residues are indicated by orange dashed lines, and those unique to Pro6 and Pro10 are indicated by green dashed lines.

The online version of this article includes the following figure supplement(s) for figure 3:

**Figure supplement 1.** Experimental details of mAb 3D11 binding.

**Figure supplement 2.** Interactions between mAb 3D11 aromatic side chains and PbCSP peptides.

## Multiple copies of mAb 3D11 bind PbCSP with high affinity

Next, we investigated the binding of mAb 3D11 to the PbCSP repeat of low structural propensity. Our biolayer interferometry (BLI) studies indicated that 3D11 Fab binds PbCSP with complex kinetics, but overall high affinity (*Figure 2A*). Isothermal titration calorimetry (ITC) also indicated a high affinity interaction, with a $K_D$ value of 159 ± 47 nM (*Figure 2B*). In addition, ITC revealed a very high binding stoichiometry (N = 10 ± 1), suggesting that approximately ten copies of 3D11 Fab bound one molecule of PbCSP simultaneously. Size-exclusion chromatography coupled with multi-angle light scattering (SEC-MALS) characterization of the 3D11 Fab-PbCSP complex confirmed the high binding stoichiometry with a molecular weight of 587 ± 7 kDa for the complex (*Figure 2C–D*). This size is consistent with approximately eleven 3D11 Fabs bound to one molecule of PbCSP, and thus, is in agreement with the results from the ITC studies within experimental error. Therefore, through a number of biophysical studies, we show that up to eleven copies of 3D11 Fab can bind simultaneously to PbCSP with high affinity.

**Table 1.** X-ray crystallography data collection and refinement statistics.

Despite binding in nearly identical conformations, differences exist in the molecular details of 3D11 Fab binding to each peptide that provide key insights into mAb 3D11 recognition of PbCSP. Our crystal structures revealed that more van der Waals contacts were formed by a Pro residue in the PPPP and NPND motifs compared to an Ala at the same position in the PAPP and NAND motifs (**Figure 3D**). Consequently, the epitopes of the NAND, NPND and Mixed peptides had a slightly greater buried surface area (BSA; 753, 762, and 765 Å$^2$, respectively) than the PAPP peptide (743 Å$^2$), which only consists of Ala-containing motifs (**Supplementary file 2**). In particular, Pro10 of the PPPP motif found in the NAND and NPND peptides forms more van der Waals interactions with antibody residues H.Asn33 and H.Tyr52 compared to Ala10 of the PAPP motif present in PAPP and Mixed peptides. Similarly, Pro6 of the NPND motif in the NPND and Mixed peptides makes additional interactions with antibody residue K.Leu50 that are not present for Ala6 of the NAND motif within the PAPP and NAND peptides (**Supplementary file 2**). These differences in interactions observed at the atomic level directly relate to the binding affinities measured by ITC, where the PbCSP peptides that bury more surface area in the 3D11 paratope have the highest binding affinities (**Figure 3A**).

| | 3D11-PAPP | 3D11-NAND | 3D11-NPND | 3D11-Mixed |
|---|---|---|---|---|
| Beamline | APS-23-ID-D | APS-23-ID-D | NSLS-II-17-ID-1 | APS-23-ID-B |
| Wavelength (Å) | 1.033170 | 1.033200 | 0.979329 | 1.033167 |
| Space group | P3$_2$21 | P3$_2$21 | P3$_2$21 | P3$_2$21 |
| Cell dimensions | | | | |
| $a,b,c$ (Å) | 59.3, 59.3, 233.5 | 59.7, 59.7, 234.9 | 59.9, 59.9, 235.0 | 60.3, 60.3, 233.7 |
| α, β, γ (°) | 90, 90, 120 | 90, 90, 120 | 90, 90, 120 | 90, 90, 120 |
| Resolution (Å)[*] | 40.0–1.60 (1.70–1.60) | 40.0–1.55 (1.65–1.55) | 40.0–2.27 (2.37–2.27) | 40.0–1.55 (1.65–1.55) |
| No. molecules in ASU | 1 | 1 | 1 | 1 |
| No. observations | 1,210,903 (196,555) | 684,564 (117,091) | 450,057 (47,142) | 1,423,235 (247,601) |
| No. unique observations | 64,371 (10,497) | 70,664 (11,753) | 23,398 (2,556) | 72,981 (12,222) |
| Multiplicity | 18.8 (18.7) | 9.5 (9.7) | 19.1 (17.4) | 19.5 (20.3) |
| R$_{merge}$ (%)[†] | 10.3 (84.7) | 8.4 (80.1) | 13.8 (57.1) | 8.3 (78.0) |
| R$_{pim}$ (%)[‡] | 2.4 (20.1) | 2.9 (26.5) | 3.2 (13.5) | 1.9 (17.6) |
| <I/σ I> | 16.3 (1.5) | 13.8 (1.5) | 19.0 (4.1) | 19.6 (1.7) |
| CC$_{½}$ | 99.9 (68.0) | 99.9 (56.7) | 99.9 (93.5) | 99.9 (84.3) |
| Completeness (%) | 99.9 (100.0) | 98.3 (97.2) | 99.3 (94.4) | 100.0 (100.0) |
| Refinement Statistics | | | | |
| Reflections used in refinement | 64,275 | 70,660 | 23,327 | 72,843 |
| Reflections used for R-free | 1999 | 1986 | 1173 | 2000 |
| Non-hydrogen atoms | 3823 | 3915 | 3665 | 3858 |
| Macromolecule | 3411 | 3423 | 3382 | 3439 |
| Water | 384 | 380 | 259 | 359 |
| Heteroatom | 28 | 112 | 24 | 60 |
| R$_{work}$[§]/R$_{free}$[¶] | 15.9/18.8 | 16.4/18.4 | 16.6/22.2 | 16.6/18.1 |
| Rms deviations from ideality | | | | |
| Bond lengths (Å) | 0.016 | 0.010 | 0.006 | 0.011 |
| Bond angle (°) | 1.43 | 1.15 | 0.87 | 1.22 |
| Ramachandran plot | | | | |
| Favored regions (%) | 98.9 | 98.0 | 97.7 | 98.2 |
| Allowed regions (%) | 1.1 | 2.0 | 2.3 | 1.8 |
| B-factors (Å$^2$) | | | | |
| Wilson B-value | 27.1 | 24.0 | 32.0 | 26.3 |
| Average B-factors | 35.0 | 31.4 | 35.2 | 31.2 |
| Average macromolecule | 33.6 | 29.4 | 34.8 | 29.7 |
| Average heteroatom | 54.4 | 54.8 | 54.4 | 57.6 |
| Average water molecule | 46.3 | 41.9 | 38.3 | 41.2 |

\* Values in parentheses refer to the highest resolution bin.

† $R_{merge} = \Sigma hkl\ \Sigma i\ |\ Ihkl, i - <Ihkl> |\ /\ \Sigma hkl <Ihkl>$ .

‡ $R_{pim} = \Sigma hkl\ [1/(N-1)]1/2\ \Sigma i\ |\ Ihkl, i - <Ihkl> |\ /\ \Sigma hkl <Ihkl>$ .

§ $R_{work} = (\Sigma\ |\ |Fo| - |Fc|\ |)\ /\ (\Sigma\ |\ |Fo|)$ - for all data except as indicated in footnote ¶.

¶ 5% of data were used for the $R_{free}$ calculation.

## mAb 3D11 is cross-reactive with subtly different PbCSP motifs in the central repeat

We next sought to define the exact mAb 3D11 epitope. We first conducted BLI studies to confirm that mAb 3D11 does not bind the PbCSP C-terminal domain (residues 202–318; *Figure 3—figure supplement 1A*). Next, we performed ITC studies to evaluate 3D11 Fab binding to each of the four peptides derived from the PbCSP central repeat region that were used in our MD simulations (*Figure 3A*). Our experiments revealed that mAb 3D11 preferentially binds the NPND and Mixed peptides with high affinity ($K_D$ = 45 ± 15 nM and 44 ± 4 nM, respectively), but also binds the NAND and PAPP peptides, albeit with lower affinity ($K_D$ = 207 ± 1 nM and 611 ± 139 nM, respectively).

To gain insight into the molecular basis of this preference, we solved the X-ray crystal structures of 3D11 Fab in complex with each peptide. The structure of the 3D11 Fab-NPND complex was determined at 2.30 Å resolution, while the structures of 3D11 Fab in complex with each of the other three peptides were all solved at ~1.60 Å resolution (*Table 1*). Interestingly, all four peptides adopted almost identical conformations when bound by 3D11 Fab (*Figure 3B* and *Figure 3—figure supplement 1*), fitting deep into the binding groove and forming a curved, U-shaped structure (*Figure 3C*). Amongst all four peptides, the mAb 3D11 core epitope consisted of eight residues [PN (A/P)NDP(A/P)P] with an all-atom RMSD <0.5 Å. Importantly, this shared recognition mode ideally positions aromatic side chains in the mAb 3D11 complementarity determining regions (CDRs) to form favorable pi-stacking and hydrophobic cage interactions around each PbCSP peptide (*Figure 3—figure supplement 2*). Indeed, the majority of these contacts are made with residues that are conserved between all four PbCSP repeat peptides, and thus, contribute to the cross-reactive binding profile of mAb 3D11.

Despite binding in nearly identical conformations, differences exist in the molecular details of 3D11 Fab binding to each peptide that provide key insights into mAb 3D11 recognition of PbCSP. Our crystal structures revealed that more van der Waals contacts were formed by a Pro residue in the PPPP and NPND motifs compared to an Ala at the same position in the PAPP and NAND motifs (*Figure 3D*). Consequently, the epitopes of the NAND, NPND and Mixed peptides had a slightly greater buried surface area (BSA; 753, 762, and 765 Å$^2$, respectively) than the PAPP peptide (743 Å$^2$), which only consists of Ala-containing motifs (*Supplementary file 2*). In particular, Pro10 of the PPPP motif found in the NAND and NPND peptides forms more van der Waals interactions with antibody residues H.Asn33 and H.Tyr52 compared to Ala10 of the PAPP motif present in PAPP and Mixed peptides. Similarly, Pro6 of the NPND motif in the NPND and Mixed peptides makes additional interactions with antibody residue K.Leu50 that are not present for Ala6 of the NAND motif within the PAPP and NAND peptides (*Supplementary file 2*). These differences in interactions observed at the atomic level directly relate to the binding affinities measured by ITC, where the PbCSP peptides that bury more surface area in the 3D11 paratope have the highest binding affinities (*Figure 3A*).

## 3D11 binding stabilizes the central PbCSP repeat in a spiral-like conformation

To understand how mAb 3D11 recognizes full-length PbCSP, we performed cryoEM analysis on the SEC-purified 3D11 Fab-PbCSP complex (*Figure 2D*). A dataset of 165,747 3D11 Fab-PbCSP particle images was refined with no symmetry imposed, resulting in a 3.2 Å resolution reconstruction of 3D11 Fabs peripherally arranged around PbCSP with their variable domains clustered around a central density (*Figure 4*, *Table 2* , and *Figure 4—figure supplement 1* and *Figure 4—figure supplement 2*). Although the low-pass filtered (20 Å) cryoEM map of the 3D11 Fab-PbCSP complex contains visible density for >10 3D11 Fabs (*Figure 4—figure supplement 2F*), only the density for the seven central Fabs was strong enough to warrant building a molecular model. Indeed, 3D

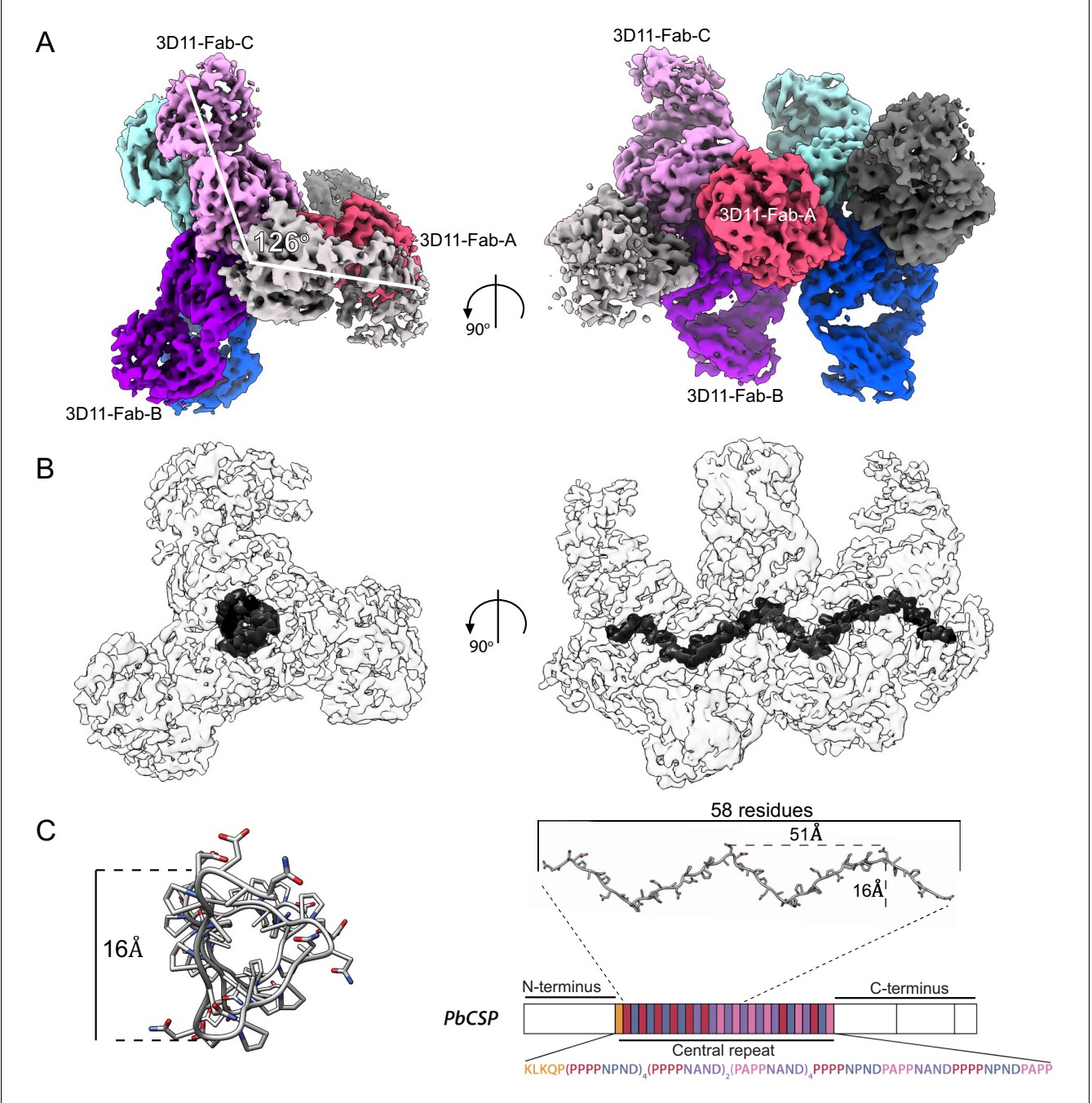

**Figure 4.** Spiral organization of the PbCSP repeat upon 3D11 Fab binding. (**A**) The cryoEM map of the 3D11 Fab-PbCSP complex reveals high-resolution information for seven predominant 3D11 Fabs. Regions corresponding to Fabs are colored from pink to gray. (**B**) CryoEM map of the 3D11 Fab-PbCSP complex is shown as a transparent light gray surface with the PbCSP region highlighted in black. (**C**) The PbCSP model built into the cryoEM map is shown in dark gray as sticks and aligned to the schematic representation of the PbCSP protein sequence.

The online version of this article includes the following video and figure supplement(s) for figure 4:

**Figure supplement 1.** CryoEM data processing workflow in cryoSPARC v2.

**Figure supplement 2.** CryoEM analysis of the 3D11 Fab-PbCSP complex.

**Figure supplement 3.** Comparison between the 3D11 Fab-PbCSP cryoEM structure and 3D11 Fab-NPND peptide crystal structure.

**Figure 4—video 1.** 3D Variability Analysis on 165,747 particle images of the 3D11 Fab-PbCSP complex.

https://elifesciences.org/articles/59018#fig4video1

Variability Analysis (*Punjani and Fleet, 2020*) in cryoSPARC v2 (*Punjani et al., 2017*) revealed continuous flexibility at the N- and C-termini of the 3D11 Fab-PbCSP complex (*Figure 4—video 1*). The PbCSP repeat forms the core of the complex and is arranged into a triangular spiral of 51 Å pitch and 16 Å diameter (*Figure 4B–C*), which fits 61 of the 108 residues in the PbCSP central region. We assigned the density to the high-affinity PPPPNPND repeats.

The angle between two Fab variable domains is ~126°, such that approximately three Fabs are required to complete one full turn of the spiral (*Figure 4A*). The cryoEM structure of the 3D11 Fab-PbCSP complex and the crystal structures of the 3D11 Fab-peptide complexes are in remarkable agreement for both the Fab (backbone RMSD = 0.69 Å) and the PbCSP repeat region (backbone RMSD = 0.66 Å; *Figure 4—figure supplement 3*). Minor differences exist in the N- and C-termini of the peptides, presumably because the termini are largely unrestricted in the crystal structures compared to the cryoEM structure.

## Contacts between 3D11 Fabs stabilize the PbCSP spiral structure

To access their repeating and densely-packed epitopes, 3D11 Fabs are closely arranged against one another in the 3D11 Fab-PbCSP complex. Indeed, the epitope for a single Fab can be defined by 14 residues (PPPPNPNDPPPPNP, *Supplementary file 3*), with the six C-terminal residues constituting the beginning of the epitope for the adjacent Fab. When considering two adjacent Fabs as a single binding unit, the BSA of the Fabs is 1313 $Å^2$, and 1636 $Å^2$ for PbCSP. Interestingly, we observe

**Table 2.** CryoEM data collection and refinement statistics.

| Data collection | |
| --- | --- |
| Electron microscope | Titan Krios G3 |
| Camera | Falcon 3EC |
| Voltage (kV) | 300 |
| Nominal magnification | 75,000 |
| Calibrated physical pixel size (Å) | 1.06 |
| Total exposure (e- /$Å^2$) | 42.7 |
| Number of frames | 30 |
| **Image processing** | |
| Motion correction software | cryoSPARCv2 |
| CTF estimation software | cryoSPARCv2 |
| Particle selection software | cryoSPARCv2 |
| 3D map classification and refinement software | cryoSPARCv2 |
| Micrographs used | 2080 |
| Particles selected | 669,223 |
| Global resolution (Å) | 3.2 |
| Particles contributing to final map | 165,747 |
| **Model building** | |
| Modeling software | Coot, phenix.real_space_refine |
| Number of residues built | 3085 |
| RMS (bonds) | 0.002 |
| RMS (angles) | 0.56 |
| Ramachandran favored (%) | 95.8 |
| Rotamer outliers (%) | 0.5 |
| Clashscore | 6.27 |
| MolProbity score | 1.63 |
| EMRinger score | 2.54 |

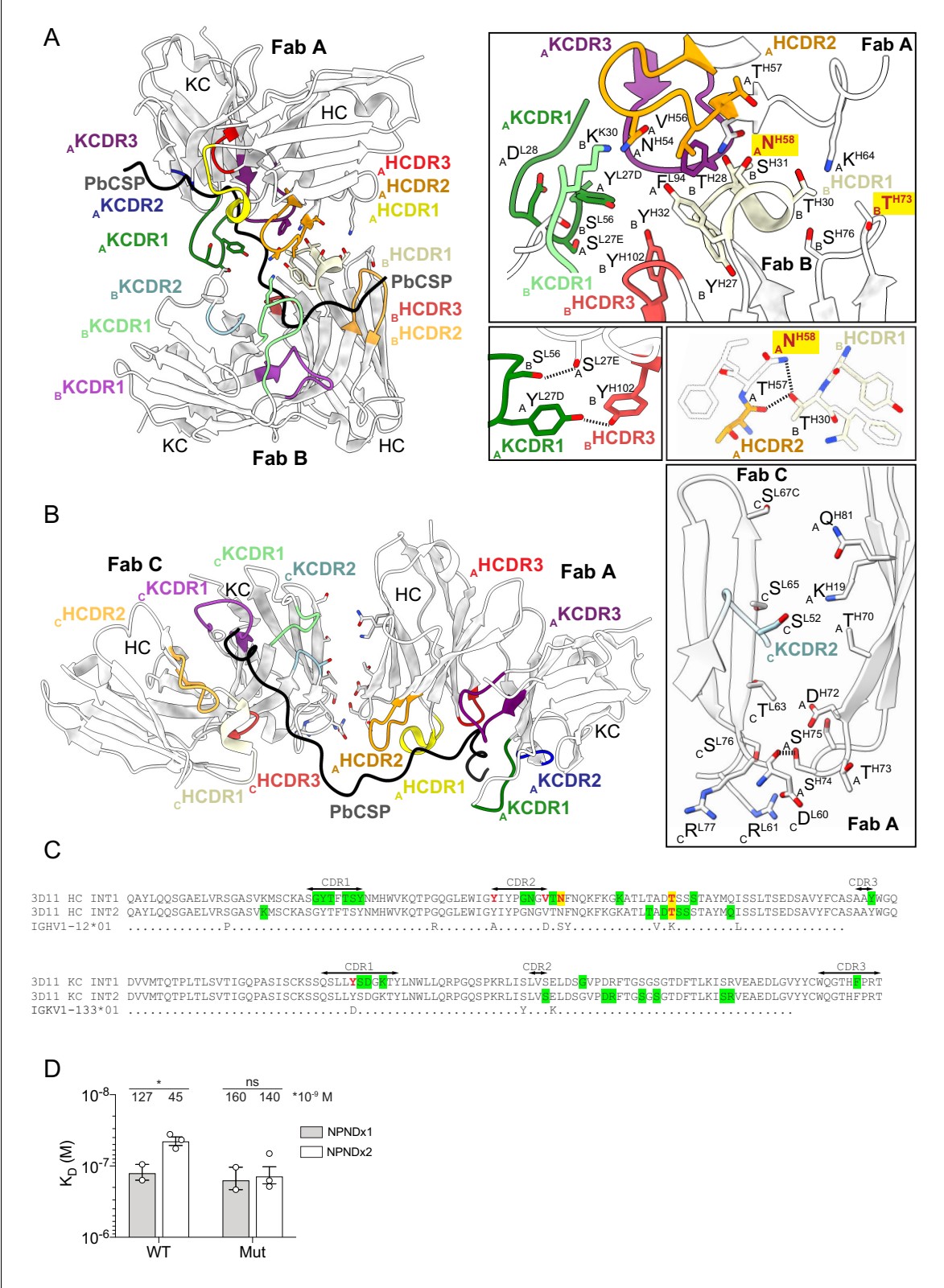

**Figure 5.** Homotypic interactions between 3D11 Fabs stabilize the 3D11 Fab-PbCSP complex. (**A** and **B**) Close-up views of adjacent 3D11 Fabs from the cryoEM structure in complex with PbCSP (black). 3D11 Fabs bound to PbCSP form homotypic contacts with each adjacent Fab through two interfaces; one consisting of CDRs from the heavy and light chains of Fabs A and B (interface 1, **A**), and the second mediated by residues in FR3 of Fab A HC and FR3 of Fab C LC (interface 2, **B**). Variable domains of Fabs are shown in white. HCDR1, −2,−3, and KCDR1, −2 and −3 are colored yellow, orange, red,

*Figure 5 continued on next page*

*Figure 5 continued*

green, blue and purple, respectively. Residues forming Fab-Fab contacts are labeled with the position of the Fab in the cryoEM model (A, B or C) indicated in subscript. mAb 3D11 affinity-matured residues that engage in Fab-Fab contacts, but do not directly interact with PbCSP are highlighted in yellow with red font. Black dashed lines denote H-bonds. (C) Sequence alignment of mAb 3D11 with its inferred germline precursor. INT1 and INT2 refer to the two interfaces shown in (A) and (B). Green highlight: germline-encoded residues involved in homotypic interactions; Red: affinity-matured residues involved in homotypic interactions; Yellow highlight: affinity-matured residues involved in homotypic interactions that do not directly interact with PbCSP. (D) Binding affinity of WT 3D11 and H-58/73 germline-reverted mutant (Mut) Fabs to NPNDx1 (gray bars) and NPNDx2 (white bars) peptides as measured by ITC. Symbols represent independent measurements. Mean $K_D$ values resulting from at least two independent experiments are shown. Error bars represent standard error of the mean. An unpaired one-tailed t-test was performed using GraphPad Prism 8 to evaluate statistical significance: *p<0.05.

The online version of this article includes the following figure supplement(s) for figure 5:

**Figure supplement 1.** Homotypic contacts between 3D11 Fabs in the 3D11 Fab-PbCSP cryoEM structure.
**Figure supplement 2.** Negative-stain EM analysis of 3D11 IgG-PbCSP complexes.
**Figure supplement 3.** Comparison between cryoEM structures of 3D11 Fab-PbCSP and 311 Fab-PfCSP (PDB ID: 6MB3) (*Oyen et al., 2018*).

multiple Fab-Fab contacts in the cryoEM structure (*Figure 5* and *Figure 5—figure supplement 1*). Comparison of the mAb 3D11 sequence to its inferred germline precursor (IGHV1-12 and IGKV1-135) reveals that some of the residues involved in these homotypic contacts have been somatically hypermutated (H.Tyr50 and H.Val56 in HCDR2, H.Asn58 and H.Thr73 in heavy chain (HC) framework region (FR) 3, and K.Tyr27D in KCDR1; *Figure 5C*). While H.Tyr50, H.Val56 and K.Tyr27D mediate Fab-Fab contacts in addition to directly interacting with PbCSP, H.Asn58 and H.Thr73 are only involved in Fab-Fab interactions.

To investigate the role of affinity maturation in enhancing Fab-Fab contacts, somatically mutated HC residues H.Asn58 and H.Thr73 were reverted to their inferred germline precursors (N58S and T73K: subsequently named H-58/73; *Figure 5C*). We performed ITC studies to evaluate binding of wild-type (WT) and H-58/73 germline-reverted mutant 3D11 Fabs to two peptides derived from PbCSP, designed based on our X-ray and cryoEM structures to constitute the minimal binding site for one 3D11 Fab (PPPPNPNDPPPP, denoted 'NPNDx1') or two 3D11 Fabs in a 'head-to-head' conformation (PPPPNPNDPPPPNPNDPPPPNPND, denoted 'NPNDx2'). Although both WT and H-58/73 germline-reverted mutant Fabs bound NPNDx1 with comparable affinity, WT 3D11 Fab demonstrated significantly greater affinity for NPNDx2 compared to NPNDx1 (KD values of 45 ± 6 nM and 127 ± 32 nM, respectively; *Figure 5D*). On the other hand, the H-58/73 germline-reverted mutant bound each peptide with similar affinities (KD values of 140 ± 38 nM for NPNDx2 and 160 ± 56 nM for NPNDx1; *Figure 5D*). The improved binding affinity of WT 3D11 Fab for NPNDx2 compared to NPNDx1, which is not observed for the H-58/73 germline-reverted mutant 3D11 Fab, suggests an important role for residues H.Asn58 and H.Thr73 in mediating homotypic interactions between neighboring 3D11 Fabs bound to PbCSP. Together, these data provide evidence for the affinity maturation of homotypic contacts that indirectly strengthen mAb 3D11 affinity to PbCSP.

To examine whether 3D11 IgG can induce a similar type of spiral conformation of PbCSP as 3D11 Fab, we prepared complexes of 3D11 IgG-PbCSP for negative-stain (ns) EM analysis. Incubation of PbCSP with excess 3D11 IgG resulted in significant precipitation of the sample, presumably due to IgG-induced crosslinking of PbCSP molecules (*Figure 5—figure supplement 2A–B*). Nonetheless, a minor soluble fraction of the complex could be purified. Comparison of nsEM 2D class averages of this 3D11 IgG-PbCSP fraction to 2D class averages of the 3D11 Fab-PbCSP complex indicated that binding of either the 3D11 Fab or IgG can induce structural ordering of PbCSP into similar spiral conformations (*Figure 5—figure supplement 2C*). Our findings are in agreement with a similar analysis previously performed with human 311 Fab and IgG in complex with PfCSP (*Oyen et al., 2018*), which also observed the ability of both IgG and Fab to induce a spiral-like conformation in CSP.

## Discussion

The CSP repeat is of broad interest for malaria vaccine design because it is targeted by inhibitory antibodies capable of preventing sporozoite infection as the parasite transits from *Anopheles* mosquitoes to mammalian hosts. Biophysical studies of the PfCSP central NANP repeat have shown that

this region possesses low secondary structure propensities (*Dyson et al., 1990*), and AFM studies on live Pf sporozoites suggest a range of conformations for PfCSP (*Patra et al., 2017*; *Herrera et al., 2015*). Importantly, recent studies have uncovered that some of the most potent antibodies against the PfCSP repeat region are cross-reactive with the PfCSP N-terminal junction, which harbors KQPA, NPDP and NVDP motifs interspersed with NANP motifs (*Kisalu et al., 2018*; *Murugan et al., 2020*; *Tan et al., 2018*). Our MD simulations of different sub-regions of the PfCSP central repeat, including the N-junction, provided detailed descriptions of their conformational ensemble and revealed that each sequence motif possesses a similarly low structural propensity.

Our MD simulations for PbCSP also indicated that the low structural propensity of central repeat motifs with subtle sequence variance extends to other *Plasmodium* species. These findings are in agreement with studies linking repetitive, low-complexity peptide sequences to structural disorder (*Rauscher and Pomès, 2012*; *Rauscher and Pomès, 2017*; *Romero et al., 2001*). The role of the numerous repetitive sequences observed in parasitic genomes (*Tan et al., 2010*; *Mendes et al., 2013*; *Davies et al., 2017*) remains to be fully understood, but is postulated to include maximizing parasite interactions with the target host cell (*Mendes et al., 2013*), allowing the parasite to adapt under selective pressure by varying its number of repeats (*Davies et al., 2017*), and impairing the host immune response (*Ly and Hansen, 2019*; *Portugal et al., 2015*; *Sullivan et al., 2015*).

Binding of the PfCSP repeat by inhibitory antibodies has been shown to induce various conformations in this intrinsically disordered region (*Imkeller et al., 2018*; *Kisalu et al., 2018*; *Murugan et al., 2020*; *Oyen et al., 2017*; *Tan et al., 2018*; *Triller et al., 2017*; *Scally and Julien, 2018*; *Pholcharee et al., 2020*). Here, we show that the PbCSP repeat adopts an extended and bent conformation when recognized by inhibitory mAb 3D11. Antibody recognition of the PfCSP repeat is often mediated by aromatic cages formed by the paratope, which surround prolines, backbone atoms, and aliphatic portions of side chains in the epitope (*Murugan et al., 2020*; *Pholcharee et al., 2020*). Antibody paratope residues partaking in aromatic cages often include germline-encoded residues, such as H.Trp52 from VH3-33 signature genes that are strongly recruited in the humoral response against PfCSP (*Julien and Wardemann, 2019*; *Pholcharee et al., 2020*; *Murugan et al., 2018*). Similarly, murine mAb 3D11 uses eight aromatic residues to recognize the PbCSP repeat. Germline-encoded K.Tyr32 appears to play a central role in mAb 3D11 PbCSP recognition by contacting consecutive Asn-Asp-Pro residues (PN(A/P)NDP(A/P)P) in the middle of the core epitope, contributing 58 Å$^2$ of BSA on the Fab. These findings indicate a central role for germline-encoded aromatic residues in antibody binding of Plasmodium CSP repeats across species.

Our structural and biophysical data demonstrated that mAb 3D11 is cross-reactive and binds the different repeat motifs of PbCSP in nearly identical conformations. Such cross-reactivity for the repeat motifs of subtle differences in PfCSP is also exhibited by inhibitory human antibodies encoded by a variety of Ig-gene combinations (*Kisalu et al., 2018*; *Murugan et al., 2020*; *Tan et al., 2018*; *Triller et al., 2017*; *Scally et al., 2018*). Notably, the inferred germline precursor genes of mAb 3D11 (IGHV1-12/IGKV1-135) share the most sequence similarity with the human IGHV1-3/IGKV2-30 genes (68% and 82% sequence identity, respectively); IGHV1-3 is the inferred germline precursor of the potent, cross-reactive human mAb CIS43 (*Kisalu et al., 2018*). Moreover, it was previously reported that human anti-PfCSP antibody affinity is often directly associated with epitope cross-reactivity (*Murugan et al., 2020*). While mAb 3D11 provides one such example in mice, further investigation is needed to determine whether favorable selection of cross-reactive clones during B cell maturation has evolved as a common mechanism of the immune response in mammals against Plasmodium CSP.

Most residues that mediate mAb 3D11 contacts with the PbCSP repeat are germline-encoded; indeed, of nine affinity-matured residues in the HC and three in the KC, only three are involved in direct contacts with the antigen (H.Trp50, H.Val56 and K.Tyr27D). Due to the repetitive nature of the central repeat motifs, multiple antibodies bind simultaneously to one CSP protein and neighboring Fabs engage in homotypic interactions (*Imkeller et al., 2018*; *Oyen et al., 2018*). Our data suggest that somatic mutations of residues that partake in Fab-Fab contacts enhance homotypic interactions and indirectly improve the binding affinity of the mAb to CSP. In this respect, mAb 3D11 recognition of PbCSP resembles binding of some neutralizing human mAbs to PfCSP (*Imkeller et al., 2018*; *Murugan et al., 2020*; *Oyen et al., 2018*). In human mAbs 311 (*Oyen et al., 2018*) and 1210 (*Imkeller et al., 2018*), CDR3 regions of both heavy and light chains appear to play a considerable role in forming Fab-Fab contacts. Interestingly, in the case of mAb 3D11, homotypic interactions are

mainly mediated by residues localized in HCDR1 and −2, KCDR1, and FR3 regions of both the HC and KC, with little contribution from residues in the CDR3 regions (with the exception of H.Tyr97 in HCDR3 and K.Phe94 in KCDR3). Taken together, these findings indicate that homotypic interactions are a feature by which the mammalian immune system can robustly engage repetitive Plasmodium antigens with high affinity in various ways. Interestingly, recent studies have reported that Fab-Fab interactions occur in other antibody-antigen complexes, providing evidence that homotypic contacts can drive diverse biology: for example, homotypic interactions were found between two nanobodies bound to a pentameric antigen (*Bernedo-Navarro et al., 2018*), and between two Rituximab antibodies bound to B cell membrane protein CD20 (*Rougé et al., 2020*).

Our cryoEM analysis also revealed how the PbCSP repeat, like that of PfCSP, can adopt a highly organized spiral structure upon mAb binding. Such spiral assembly of CSP was previously observed upon human mAb 311 Fab and IgG binding, which induced a PfCSP spiral with a greater diameter (27 Å) and smaller pitch (49 Å) compared to the 3D11-PbCSP complex (16 Å diameter and 51 Å pitch) (*Oyen et al., 2018*; *Figure 5—figure supplement 3*). Differences in the architecture between these two complexes can be attributed to the fact that mAbs 3D11 and 311 recognize their respective antigens in distinct conformations. Because different anti-CSP inhibitory antibodies can bind the repeat region in a variety of conformations (*Imkeller et al., 2018*; *Kisalu et al., 2018*; *Tan et al., 2018*; *Triller et al., 2017*; *Scally and Julien, 2018*), it is likely that many types of CSP-antibody assemblies exist. Further studies are needed to investigate whether the formation of such highly organized complexes is possible on the surface of live sporozoites and how antibody-CSP interactions occur in the context of polyclonal serum. These insights will be important for our structure-function understanding of the mechanisms employed by these repeat-targeting antibodies to inhibit sporozoite development, migration and infection of hepatocytes.

# Materials and methods

## Key resources table

| Reagent type (species) or resource | Designation | Source or reference | Identifiers | Additional information |
|---|---|---|---|---|
| Recombinant DNA reagent | pcDNA3.4-3D11 Fab HC (plasmid) | This paper | N/A | 3D11 Fab heavy chain gene in pcDNA3.4 TOPO vector |
| Recombinant DNA reagent | pcDNA3.4-3D11 Fab 58/73 HC (plasmid) | This paper | N/A | 3D11 Fab germline-reverted mutant heavy chain gene in pcDNA3.4 TOPO vector |
| Recombinant DNA reagent | pcDNA3.4-3D11 Fab KC (plasmid) | This paper | N/A | 3D11 Fab light chain gene in pcDNA3.4 TOPO vector |
| Recombinant DNA reagent | pcDNA3.4-PbCSP-6xHis (plasmid) | This paper | N/A | PbCSP gene with His tag in pcDNA3.4 TOPO vector |
| Recombinant DNA reagent | pcDNA3.4-PbC-CSP-6xHis (plasmid) | This paper | N/A | PbC-CSP gene with His tag in pcDNA3.4 TOPO vector |
| Recombinant DNA reagent | pcDNA3.4-PbCSP-αTSR-6xHis (plasmid) | This paper | N/A | PbCSP αTSR gene with His tag in pcDNA3.4 TOPO vector |
| Cell line (*Homo sapiens*) | FreeStyle 293 F cells | Thermo Fisher Scientific | Cat# R79007 | |
| Cell line (*Mus musculus*) | 3D11 hybridoma cell line | *Yoshida et al., 1980* | BEI Resources #MRA-100; RRID:AB_2650479 | |
| Chemical compound | GIBCO FreeStyle 293 Expression Medium | Thermo Fisher Scientific | Cat# 12338026 | |
| Chemical compound | GIBCO Hybridoma-SFM | Thermo Fisher Scientific | Cat# 12045076 | |
| Chemical compound | FectoPRO DNA Transfection Reagent | VWR | Cat# 10118–444 | |
| Chemical compound | Fetal bovine serum | Thermo Fisher Scientific | Cat# 12483–020 | |
| Antibody | 3D11 IgG (mouse monoclonal) | *Yoshida et al., 1980* | N/A | Purified from 3D11 hybridoma cell line; See Materials and methods |

*Continued on next page*

Continued

| Reagent type (species) or resource | Designation | Source or reference | Identifiers | Additional information |
|---|---|---|---|---|
| Recombinant protein | 3D11 Fab | This paper | N/A | See Materials and methods for concentrations and masses used, and buffer conditions |
| Recombinant protein | 3D11 Fab H-58/73 | This paper | N/A | See Materials and methods for concentrations and masses used, and buffer conditions |
| Recombinant protein | PbCSP | This paper | N/A | See Materials and methods for concentrations and masses used, and buffer conditions |
| Peptide | PAPP (PAPPNANDPAPPNAND) | This paper | N/A | Derived from PbCSP repeat region |
| Peptide | NAND (PPPPNANDPPPPNAND) | This paper | N/A | Derived from PbCSP repeat region |
| Peptide | NPND (PPPPNPNDPPPPNPND) | This paper | N/A | Derived from PbCSP repeat region |
| Peptide | Mixed (PPPPNPNDPAPPNAND) | This paper | N/A | Derived from PbCSP repeat region |
| Peptide | NPNDx1 (PPPPNPNDPPPP) | This paper | N/A | Derived from PbCSP repeat region |
| Peptide | NPNDx2 (PPPPNPNDPPPP NPNDPPPPNPND) | This paper | N/A | Derived from PbCSP repeat region |
| Software, algorithm | GROMACS 5.1.4 | *Abraham et al., 2015*; *Berendsen et al., 1995* | http://manual.gromacs.org/ documentation/5.1.4/; RRID:SCR_014565 | |
| Software, algorithm | LINCS | *Hess et al., 1997*; *Hess, 2008* | N/A | |
| Software, algorithm | Particle-Mesh Ewald algorithm | *Darden et al., 1993*; *Essmann et al., 1995* | N/A | |
| Software, algorithm | Nosé-Hoover thermostat | *Nosé, 1984*; *Hoover, 1985* | N/A | |
| Software, algorithm | Parrinello-Rahman algorithm | *Parrinello and Rahman, 1981* | N/A | |
| Software, algorithm | VMD | *Humphrey et al., 1996* | https://www.ks.uiuc.edu/ Research/vmd/; RRID:SCR_001820 | |
| Software, algorithm | Matplotlib | *Hunter, 2007* | https://matplotlib.org/; RRID:SCR_008624 | |
| Software, algorithm | Octet Data Analysis Software 9.0.0.6 | ForteBio | https://www.fortebio.com/ products/octet-systems-software | |
| Software, algorithm | MicroCal ITC Origin 7.0 Analysis Software | Malvern | https://www.malvern panalytical.com/ | |
| Software, algorithm | ASTRA | Wyatt | https://www.wyatt.com/ products/software/astra.html; RRID:SCR_016255 | |
| Software, algorithm | GraphPad Prism 8 | GraphPad Software | https://www.graphpad.com/; RRID:SCR_002798 | |
| Software, algorithm | EPU | ThermoFisher Scientific | https://www.fei.com/software/ | |
| Software, algorithm | SBGrid | SBGrid Consortium | https://sbgrid.org/; RRID:SCR_003511 | |
| Software, algorithm | cryoSPARC v2 | *Punjani et al., 2017* | https://cryosparc.com/ | |
| Software, algorithm | Phenix (phenix.refine; phenix.real_space_refine) | *Adams et al., 2010* | https://www.phenix-online.org/; RRID:SCR_014224 | |

*Continued on next page*

*Continued*

| Reagent type (species) or resource | Designation | Source or reference | Identifiers | Additional information |
|---|---|---|---|---|
| Software, algorithm | UCSF Chimera | *Pettersen et al., 2004* | https://www.cgl.ucsf.edu/chimera/; RRID:SCR_004097 | |
| Software, algorithm | UCSF ChimeraX | *Goddard et al., 2018* | https://www.cgl.ucsf.edu/chimerax/; RRID:SCR_015872 | |
| Software, algorithm | Coot | *Emsley et al., 2010* | https://www2.mrc-lmb.cam.ac.uk/personal/pemsley/coot/; RRID:SCR_014222 | |
| Software, algorithm | PyMOL | The PyMOL Molecular Graphics System, Version 1.8 Schrödinger, LLC. | https://pymol.org/2/#products; RRID:SCR_000305 | |
| Software, algorithm | PDBePISA | *Krissinel and Henrick, 2007* | https://www.ebi.ac.uk/pdbe/pisa/; RRID:SCR_015749 | |
| Other | Homemade holey gold grids | *Marr et al., 2014* | N/A | |
| Other | Homemade carbon grids | *Booth et al., 2011* | N/A | |

## Molecular dynamics simulations

We performed all-atom molecular dynamics simulations of the following peptides: $(NPNA)_5$, K QPADGNPDPNANPN, NPDPNANPNVDPNANP, $(NVDPNANP)_2NVDP$, $(PPPPNPND)_2$, $(PPPPNAND)_2$, $(PAPPNAND)_2$, and PPPPNPNDPAPPNAND as blocked monomers in water with 0.15 M NaCl. Each simulation system consisted of the respective peptide with an acetylated N-terminus and amidated C-terminus solvated in a dodecahedral box with side lengths of 4.9 nm.

The systems were simulated using the program GROMACS 5.1.4 (67, 68) with the CHARMM22* (*Piana et al., 2011*; *Best and Hummer, 2009*; *Lindorff-Larsen et al., 2012*; *Best and Mittal, 2010*; *MacKerell et al., 1998*) force field for the protein and the TIP3P (*Jorgensen et al., 1983*) water model. All simulations were performed with periodic boundary conditions at a constant pressure and temperature of 1 bar and 300 K, respectively. The LINCS algorithm was used to constrain all bond lengths (*Hess et al., 1997*; *Hess, 2008*). A cut-off of 1.4 nm was used for Lennard-Jones interactions. The Particle-Mesh Ewald algorithm (*Darden et al., 1993*; *Essmann et al., 1995*) was used to calculate long-range electrostatics interactions with a Fourier spacing of 0.12 and an interpolation order of 4. The Nosé-Hoover thermostat (*Nosé, 1984*; *Hoover, 1985*) was used for temperature coupling with the peptide and solvent coupled to two temperature baths and a time constant of 0.1 ps. The Parrinello-Rahman algorithm (*Parrinello and Rahman, 1981*) was used for pressure coupling with a time constant of 2 ps. The integration step size was two fs and the system coordinates were stored every 10 ps.

The simulations were performed for 300 ns for 20 independent replicas of $(NPNA)_5$ and 10 independent replicas of all other sequences. The initial structures of the peptides were selected from 10 ns simulations in which extended conformations of the peptides were collapsed *in vacuo*. The first 100 ns of each trajectory were omitted as the time required for system relaxation based on the convergence analysis of the radius of gyration ($R_g$) shown in *Figure 1—figure supplement 2*. This protocol resulted in a total of 4 µs of production time for the $(NPNA)_5$ dataset and a total of 2 µs of production time for the other systems, which was used to compute equilibrium ensemble properties. The peptide snapshots were generated with VMD (*Humphrey et al., 1996*) and the plots were created with Matplotlib (*Hunter, 2007*).

## 3D11 Fab production and purification

The mAb 3D11 hybridoma cell line variable light and heavy chain antibody genes were sequenced (Applied Biological Materials Inc). mAb 3D11 $V_K$ and $V_H$ regions were cloned individually into custom pcDNA3.4 expression vectors immediately upstream of human Igκ and Igγ1-$C_H$1 domains,

respectively. The resulting pcDNA3.4-3D11 Fab KC and −3D11 Fab HC or −3D11 Fab 58/73 HC plasmids were co-transfected into FreeStyle 293 F cells for transient expression using FectoPRO DNA Transfection Reagent, cultured in GIBCO FreeStyle 293 Expression Medium, and purified via KappaSelect affinity chromatography (GE Healthcare), cation exchange chromatography (MonoS, GE Healthcare), and size-exclusion chromatography (Superdex 200 Increase 10/300 GL, GE Healthcare).

## 3D11 IgG production and purification

The mAb 3D11 hybridoma cell line (BEI Resources MRA-100) was cultured in GIBCO Hybridoma-SFM (Thermo Fisher Scientific Cat#12045076) with 2.5–10% fetal bovine serum (Thermo Fisher Scientific Cat#12483–020). Cells were harvested and the supernatant containing 3D11 IgG was purified via Protein G affinity chromatography (GE Healthcare) and size-exclusion chromatography (Superose 6 Increase 10/300 GL, GE Healthcare).

## Recombinant PbCSP production and purification

Constructs of full-length PbCSP (residues 24–318), the PbCSP C-terminal domain (residues 202–318; PbC-CSP) and the PbCSP αTSR domain (residues 263–318; PbCSP αTSR) from strain ANKA (NCBI reference sequence XP_022712148.1) were designed with potential N-linked glycosylation sites mutated to glutamine and cloned into pcDNA3.4 expression vectors with a His tag. The resulting pcDNA3.4-PbCSP-6xHis, -PbC-CSP-6xHis and -PbCSP-αTSR-6xHis plasmids were transiently transfected in FreeStyle 293 F cells using FectoPRO DNA Transfection Reagent, cultured in GIBCO FreeStyle 293 Expression Medium, and purified by HisTrap FF affinity chromatography (GE Healthcare) and size-exclusion chromatography (Superdex 200 Increase 10/300 GL, GE Healthcare).

## Cell lines

FreeStyle 293 F cells (Thermo Fisher Scientific 12338026) and 3D11 hybridoma cell line (BEI Resources MRA-100) were authenticated and validated to be mycoplasma-free by their respective commercial entities.

## Binding kinetics by biolayer interferometry

BLI (Octet RED96, FortéBio) experiments were conducted to determine the binding kinetics of the 3D11 Fab to recombinant PbCSP. PbCSP, PbC-CSP or PbCSP αTSR was diluted to 10 µg/ml in kinetics buffer (PBS, pH 7.4, 0.01% [w/v] BSA, 0.002% [v/v] Tween-20) and immobilized onto Ni-NTA (NTA) biosensors (FortéBio). After a steady baseline was established, biosensors were dipped into wells containing twofold dilutions of 3D11 Fab in kinetics buffer. Tips were then immersed back into kinetics buffer for measurement of the dissociation rate. Kinetics data were analyzed using the FortéBio's Octet Data Analysis software 9.0.0.6, and curves were fitted to a 2:1 binding model.

## Binding thermodynamics by isothermal titration calorimetry

Calorimetric titration experiments were performed with an Auto-iTC$_{200}$ instrument (Malvern) at 37°C. Full-length PbCSP and PbCSP-derived peptides (PAPP, NAND, NPND, Mixed, NPNDx1, NPNDx2; GenScript) were diluted in Tris-buffered saline (TBS; 20 mM Tris pH 8.0, and 150 mM NaCl) and added to the calorimetric cell. Titrations were performed with 3D11 Fab in the syringe, diluted in TBS, in 15 successive injections of 2.5 µl. Full-length PbCSP was diluted to 5 µM and titrated with 3D11 Fab at 400 µM. All PbCSP-derived peptides were diluted to 20 µM and titrated with 3D11 Fab at 200–300 µM; with the exception of the NPNDx2 peptide, which was diluted to 9–10 µM and titrated with 180–200 µM 3D11 Fab. Experiments were performed at least two times, and the mean and standard error of the mean are reported. The experimental data were analyzed using the Micro-Cal ITC Origin 7.0 Analysis Software according to a 1:1 binding model.

## Size-exclusion chromatography-multi-angle light scattering (SEC-MALS)

Full-length PbCSP was complexed with a molar excess of 3D11 Fab and loaded on a Superose 6 Increase 10/300 GL (GE Healthcare) using an Agilent Technologies 1260 Infinity II HPLC coupled in-line with the following calibrated detectors: (i) MiniDawn Treos MALS detector (Wyatt); (ii)

Quasielastic light scattering (QELS) detector (Wyatt); and (iii) Optilab T-reX refractive index (RI) detector (Wyatt). Data processing was performed using the ASTRA software (Wyatt).

## Crystallization and structure determination

Purified 3D11 Fab was concentrated and diluted to 5 mg/mL with each of the PAPP, NAND and Mixed peptides in a 1:5 molar ratio; and diluted to 2.1 mg/mL with the NPND peptide in a 1:5 molar ratio. The 3D11 Fab/PAPP complex was mixed in a 1:1 ratio with 20% (w/v) PEG 3350, 0.15 M malic acid pH 7. Crystals appeared after ~1 d and were cryoprotected in 15% (v/v) ethylene glycol before being flash-frozen in liquid nitrogen. The 3D11 Fab/NAND complex was mixed in a 1:1 ratio with 20% (w/v) PEG 3350, 0.2 M di-sodium tartrate. Crystals appeared after ~3 d and were cryoprotected in 15% (v/v) ethylene glycol before being flash-frozen in liquid nitrogen. The 3D11 Fab/NPND complex was mixed in a 1:1 ratio with 25% (w/v) PEG 3350, 0.2 M lithium sulfate, 0.1 M Tris pH 8.5. Crystals appeared after ~12 d and were cryoprotected in 20% (v/v) ethylene glycol before being flash-frozen in liquid nitrogen. The 3D11 Fab/Mixed complex was mixed in a 1:1 ratio with 25.5% (w/v) PEG 4000, 15% (v/v) glycerol, 0.17 M ammonium acetate, 0.085 M sodium citrate pH 5.6. Crystals appeared after ~1 d and were cryoprotected in 20% (v/v) glycerol before being flash-frozen in liquid nitrogen.

Data were collected at the 23-ID-D or 23-ID-B beamline at the Argonne National Laboratory Advanced Photon Source, or at the 17-ID-1 beamline at the National Synchrotron Light Source II. All datasets were processed and scaled using XDS (*Kabsch, 2010*). The structures were determined by molecular replacement using Phaser (*McCoy et al., 2007*). Refinement of the structures was performed using phenix.refine (*Adams et al., 2010*) and iterations of refinement using Coot (*Emsley et al., 2010*). Access to all software was supported through SBGrid (*Morin et al., 2013*).

## CryoEM data collection and image processing

The PbCSP/3D11 complex was concentrated to 3 mg/mL and incubated briefly with 0.01% (w/v) n-Dodecyl β-D-maltopyranoside. 3 µl of the sample was deposited on homemade holey gold grids (*Marr et al., 2014*), which were glow-discharged in air for 15 s before use. Sample was blotted for 12.5 s, and subsequently plunge-frozen in a mixture of liquid ethane and propane (*Tivol et al., 2008*) using a modified FEI Vitrobot (maintained at 4˚C and 100% humidity). Data collection was performed on a Thermo Fisher Scientific Titan Krios G3 operated at 300 kV with a Falcon 3EC camera automated with the EPU software. A nominal magnification of 75,000× (calibrated pixel size of 1.06 Å) and defocus range between 1.6 and 2.2 µm were used for data collection. Exposures were fractionated as movies of 30 frames with a total exposure of 42.7 electrons/Å$^2$. A total of 2080 raw movies were obtained.

Image processing was carried out in cryoSPARC v2 (*Punjani et al., 2017*). Initial specimen movement correction, exposure weighting, and CTF parameters estimation were done using patch-based algorithms. Manual particle selection was performed on 30 micrographs to create templates for template-based picking. 669,223 particle images were selected by template picking and individual particle images were corrected for beam-induced motion with the local motion algorithm (*Rubinstein and Brubaker, 2015*). *Ab-initio* structure determination revealed that most particles present in the dataset correspond to the 3D11 Fab-PbCSP complex, with a minor population of particles corresponding to unbound 3D11 Fab. After several rounds of heterogeneous refinement, 165,747 particle images were selected for non-uniform refinement with no symmetry applied, which resulted in a 3.2 Å resolution map of the 3D11 Fab-PbCSP complex estimated from the gold-standard Fourier shell correlation (FSC) criterion.

## CryoEM model building

To create a starting model of the 3D11 Fab-PbCSP complex, seven copies of 3D11 Fab/PbCSP-peptide crystal structures were manually docked into the 3D11 Fab-PbCSP cryoEM map using UCSF Chimera (*Pettersen et al., 2004*), followed by manual building using Coot (*Emsley et al., 2010*). All models were refined using phenix.real_space_refine (*Adams et al., 2010*) with secondary structure and geometry restraints. The final models were evaluated by MolProbity (*Chen et al., 2010*). The figures were prepared with UCSF Chimera (*Pettersen et al., 2004*) and UCSF ChimeraX

(*Goddard et al., 2018*). Contacts in the 3D11 Fab-PbCSP complex were identified by PDBePISA (*Krissinel and Henrick, 2007*).

## Negative-stain EM of 3D11 IgG-PbCSP complex

To obtain soluble complexes of 3D11-IgG-PbCSP for NS analysis, 8.4 µg of PbCSP was incubated overnight with 20x molar excess of 3D11 IgG. After removal of aggregates via centrifugation, 3D11 IgG-PbCSP complexes were purified on a Superose 6 Increase 10/300 GL column (GE Healthcare). Fractions containing complexes of 3D11 IgG-PbCSP were pooled and concentrated, and subsequently deposited at approximately 50 µg/mL onto homemade carbon grids and stained with 2% uranyl formate. Data were collected with a FEI Tecnai T20 electron microscope operating at 200 kV, and acquired with an Orius charge-coupled device (CCD) camera (Gatan Inc) at a calibrated 34,483X magnification, resulting in a pixel size of 2.71 Å. Particle picking, extraction and three rounds of 2D classification with 50 classes allowed were performed with cryoSPARC v2 (*Punjani et al., 2017*).

## Acknowledgements

We are grateful to Dr. Samir Benlekbir for help with cryoEM data collection and for advice regarding specimen preparation. We thank Dr. Stephen Scally for his input during the course of this work. This work was supported by the CIFAR Azrieli Global Scholar program (JPJ), the Ontario Early Researcher Award program (JPJ), the Canada Research Chair program (JPJ and JLR), and the Canadian Institutes of Health Research (RP). I.K. was supported by a SickKids Restracomp Fellowship, E.T. by a CIHR Canada Graduate Scholarship, and A.S. by an NSERC Canada Graduate Scholarship and a SickKids Restracomp Scholarship. This research was enabled in part by support provided by Compute Ontario (https://computeontario.ca/) and Compute Canada (https://www.computecanada.ca/). The ITC and BLI instruments were accessed at the Structural and Biophysical Core Facility, The Hospital for Sick Children, supported by the Canada Foundation for Innovation and Ontario Research Fund. CryoEM data was collected at the Toronto High Resolution High Throughput cryoEM facility, supported by the Canada Foundation for Innovation and Ontario Research Fund. X-ray diffraction experiments were performed at GM/CA@APS, which has been funded in whole or in part with federal funds from the National Cancer Institute (ACB-12002) and the National Institute of General Medical Sciences (AGM-12006). The Eiger 16M detector was funded by an NIH–Office of Research Infrastructure Programs High-End Instrumentation grant (1S10OD012289-01A1). This research used resources of the Advanced Photon Source, a U.S. Department of Energy (DOE) Office of Science user facility operated for the DOE Office of Science by Argonne National Laboratory under contract DE-AC02-06CH11357. X-ray diffraction experiments were also performed at the National Synchrotron Light Source II, a U.S. Department of Energy (DOE) Office of Science User Facility operated for the DOE Office of Science by Brookhaven National Laboratory under Contract No. DE-SC0012704. The Life Science Biomedical Technology Research resource is primarily supported by the National Institute of Health, National Institute of General Medical Sciences (NIGMS) through a Biomedical Technology Research Resource P41 grant (P41GM111244), and by the DOE Office of Biological and Environmental Research (KP1605010). The following reagent was obtained through BEI Resources, NIAID, NIH: Hybridoma 3D11 Anti-Plasmodium berghei 44-Kilodalton Sporozoite Surface Protein (Pb44), MRA-100, contributed by Victor Nussenzweig. X-ray crystallography and cryoEM data and structures are accessible from the Protein Data Bank and the Electron Microscopy Data Bank under PDB IDs 6X8P, 6X8Q, 6X8S, 6X8U and 6X87, and EMDB 22089, respectively.

## Additional information

### Funding

| Funder | Grant reference number | Author |
|---|---|---|
| Canadian Institute for Advanced Research | Azrieli Global Scholar program | Jean-Philippe Julien |
| Ontario Ministry of Economic Development, Job Creation and Trade | | John L Rubinstein<br>Jean-Philippe Julien |

| Canada Research Chairs | | John L Rubinstein Jean-Philippe Julien |
|---|---|---|
| Canadian Institutes of Health Research | | Régis Pomès |
| Canadian Institutes of Health Research | Canada Graduate Scholarship | Elaine Thai |
| Natural Sciences and Engineering Research Council of Canada | Canada Graduate Scholarship | Ananya Srivastava |
| Canada Foundation for Innovation | | John L Rubinstein Jean-Philippe Julien |
| Ontario Research Foundation | Early ResearcherAward program | Jean-Philippe Julien |
| Sick Kids Foundation | Restracomp Fellowship | Iga Kucharska |

The funders had no role in study design, data collection and interpretation, or the decision to submit the work for publication.

### Author contributions

Iga Kucharska, Formal analysis, Validation, Investigation, Visualization, Methodology, Writing - original draft, Writing - review and editing; Elaine Thai, Ananya Srivastava, Formal analysis, Funding acquisition, Validation, Investigation, Visualization, Methodology, Writing - original draft, Writing - review and editing; John L Rubinstein, Supervision, Funding acquisition, Validation, Visualization, Methodology, Writing - review and editing; Régis Pomès, Supervision, Funding acquisition, Validation, Visualization, Methodology, Writing - original draft, Writing - review and editing; Jean-Philippe Julien, Conceptualization, Supervision, Funding acquisition, Validation, Visualization, Methodology, Writing - original draft, Project administration, Writing - review and editing

### Author ORCIDs

Iga Kucharska https://orcid.org/0000-0001-6150-3419
Elaine Thai https://orcid.org/0000-0001-7576-154X
John L Rubinstein http://orcid.org/0000-0003-0566-2209
Régis Pomès http://orcid.org/0000-0003-3068-9833
Jean-Philippe Julien https://orcid.org/0000-0001-7602-3995

### Decision letter and Author response

Decision letter https://doi.org/10.7554/eLife.59018.sa1
Author response https://doi.org/10.7554/eLife.59018.sa2

---

## Additional files

### Supplementary files

• Supplementary file 1. Hydrogen-bonding propensities from simulations of peptides in solution. (**A**) Hydrogen-bonding propensity for each simulated motif and lifetime of each β-turn for the four PfCSP-derived peptides. (**B**) Hydrogen-bonding propensity for each simulated motif and lifetime of each β-turn for the four PbCSP-derived peptides.

• Supplementary file 2. Table of contacts between 3D11 Fab and PbCSP peptides. Rows are shaded according to the number of times interactions are observed between all four crystal structures, summed in the final column.

• Supplementary file 3. Table of contacts between one of the 3D11 Fabs and PbCSP in the cryoEM.

• Transparent reporting form

## Data availability

X-ray crystallography and cryoEM data and structures have been deposited to the Protein Data Bank and the Electron Microscopy Data Bank.

The following datasets were generated:

| Author(s) | Year | Dataset title | Dataset URL | Database and Identifier |
|---|---|---|---|---|
| Thai E, Julien JP | 2020 | Crystal structure of 3D11 Fab in complex with Plasmodium berghei circumsporozoite protein PAPP peptide | http://www.rcsb.org/structure/6X8Q | RCSB Protein Data Bank, 6X8Q |
| Thai E, Julien JP | 2020 | Crystal structure of 3D11 Fab in complex with Plasmodium berghei circumsporozoite protein NAND peptide | http://www.rcsb.org/structure/6X8S | RCSB Protein Data Bank, 6X8S |
| Thai E, Julien JP | 2020 | Crystal structure of 3D11 Fab in complex with Plasmodium berghei circumsporozoite protein NPND peptide | http://www.rcsb.org/structure/6X8P | RCSB Protein Data Bank, 6X8P |
| Thai E, Julien JP | 2020 | Crystal structure of 3D11 Fab in complex with Plasmodium berghei circumsporozoite protein Mixed peptide | http://www.rcsb.org/structure/6X8U | RCSB Protein Data Bank, 6X8U |
| Kucharska I, Thai E, Rubinstein J, Julien JP | 2020 | CryoEM structure of the Plasmodium berghei circumsporozoite protein in complex with inhibitory mouse antibody 3D11 | http://www.rcsb.org/structure/6X87 | RCSB Protein Data Bank, 6X87 |
| Kucharska I, Thai E, Rubinstein J, Julien JP | 2020 | CryoEM structure of the Plasmodium berghei circumsporozoite protein in complex with inhibitory mouse antibody 3D11 | http://www.ebi.ac.uk/pdbe/entry/emdb/EMD-22089 | Electron Microscopy Data Bank, 22089 |

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
