## [Decision Letter]

**Acceptance summary:**

Reviewers and the Editor were most impressed by this integrative structural biology analysis of the interaction between a monoclonal antibody and a Plasmodium protein target. This thoughtfully designed and well executed study expands our understanding of structural correlates of antibody protection against malaria. It also illustrates how the synergy of X-ray crystallography, single-particle cryo-EM and computer simulations can be uniquely revealing in the study of complex processes in molecular biophysics.

**Decision letter after peer review:**

Thank you for submitting your article "Structural ordering of the *Plasmodium berghei* circumsporozoite protein repeats by inhibitory antibody 3D11" for consideration by *eLife*. Your article has been reviewed by three peer reviewers, and the evaluation has been overseen by José Faraldo-Gómez as the Senior and Reviewing Editor. Reviewers #1 (Wai-Hong Tham) and #3 (Andrew B Ward) have agreed to reveal their identity.

The reviewers have discussed their reviews with one another. Based on this discussion, and on the individual reviews, we are glad to be able to invite you to submit a revised version of the manuscript that addresses the questions raised by the reviewers and their requests for clarification.

Reviewer #1:

The manuscript by Kucharska and Thai et al. describes the mechanism of binding of mouse monoclonal 3D11 to its target antigen *P. berghei* circumsporozoite protein (CSP). CSP is the component of the leading malaria vaccine and previous work had used rodent models to understand the inhibitory nature of antibodies. 3D11 was originally isolated from mice exposed to irradiated *P. berghei* infected mosquitoes. While it has been used widely in many studies, the structural mechanism of binding has not been elucidated, which is the focus of this manuscript. While several studies have now shown how human antibodies bind to PfCSP, it is still important to understand how 3D11 inhibits and how it binds as a comparative study and key piece of knowledge in the development of CSP as a vaccine.

There is very little to fault in this manuscript. It employs molecular dynamics, X-ray crystallography and cryo-EM – all three complementary techniques to fully visualise how 3D11 binds a single repeat, mixed repeats and on full length PbCSP. If the paper had feature only one or two of these techniques, the mechanism of binding would have been incomplete. The employment of these structural techniques is particularly important for PbCSP, as most inhibitory antibodies bind to the central regions which contain several NANNP, PPPP, PAPP, NPND or NAND motifs so understanding the binding, in particular by cryo-EM is critical.

My only query is how does the binding change with the use of intact monoclonal antibodies? I do not need to see experimental data for this as I understand it would complicate the structural approaches. Nevertheless, either by vaccination or by antibody-based delivery, intact antibodies are binding. It would be good to hear about how this constrains the binding of repeats and if changes in the IgG backbone can be part of antibody design.

1) For example, do you observe the same number of 3D11 molecules bound to PbCSP complex to be the same if Fab or intact antibody is used using SEC-MALS?

2) Would you expect to see differences in cryo-EM if the intact mAb was used?

Reviewer #2:

In the present study, Kucharska and colleagues use an integrative structural biology approach to characterize the binding of a monoclonal antibody 3D11 to the Plasmodium circumsporozoite protein (CSP). This thoughtfully designed and executed study is straightforward in its analyses building incrementally on previous data and expands our understanding of structural correlates of antibody protection against malaria. The structural data are particularly impressive and highlight the power of combining X-ray crystallography with single particle cryoEM – the high-resolution X-ray crystal structures with CSP peptides and the 3D11 Fab provide atomic-level details of molecular interactions while the cryoEM structure informs on the overall "structural ordering" of CSP upon antibody binding.

Reviewer #3:

Kucharska and Thai et al. use x-ray crystallography and single particle cryoEM examine the molecular principles by which a protective and well-characterized mouse mAb 3D11 engages PbCSP. This work is significant because, to my knowledge, there are no structural studies of antibodies in complex with repeats from Plasmodium species other than *P. falciparum*. Further, the discovery of evolved inter-Fab contacts against PbCSP is highly compelling and suggests this is a phenomenon of CSP independent of specific repeat sequence and mammalian host. Overall, the structural data are convincing and the analysis is well-done. This paper warrants publication in *eLife* as it stands, with minor adjustments outlined below.

1) From the way it is presented, it is hard to fully appreciate the details of the cryo-EM structure of the 3d11-PbCSP complex, particularly with regards to the inter-Fab contacts. It would be helpful to describe the general locations of these residues within the Fab structure (CDRH/L 1,2,3; FR). This could be compared to the published 311-rsCSP structure in order to understand the structural properties of spiral-inducing antibodies in general. Also see comments on Figure 5 below.

2) From the binding studies, the authors conclude that 10 or 11 Fabs can bind one molecule of PbCSP. The 2D classes in Figure 4—figure supplement 2B also show what appear to be more than 7 Fabs in a complex. Did the authors use 3D classification to sort out these extended complexes, or are the "trailing" Fabs simply blurred out in the high resolution reconstruction? Did the authors see any reconstructions with a fully occupied CSP? In the Materials and methods, and/or in a supplementary figure, it would be useful to have a few more details of the data processing workflow in Cryosparc, which would inform of if and how the low affinity PPAP/NAND repeats can be occupied simultaneously with the PPPP/NPND repeats.

3) Figure 5C. It is quite difficult to understand the structural details presented here. This figure would benefit greatly from making at least two panels which present two different views of the interface which focus on different details of the inter-fab contacts. Also, importantly, there is very little structural context outside of the side-chains shown, and the CDR loops to which these residues belong to are unlabeled. As mentioned above, it would be useful to know what general structural features of these antibodies can be utilized to drive high-avidity complex formation on CSP and by consequence stabilize long-range helices.

4) Figure 4—figure supplement 3 is a little bit confusing. The authors state that the x-ray and cryoEM structures are remarkably similar, which it expected. However, as colored, there are some local changes that are 1.5-2.5 Å RMSD, which is actually fairly large. Presumably this is due to the calculation based on all atom, not backbone. I would expect the larger distances to be localized to side chains rather than backbone, which is colored. It may therefore be better to represent the structure as a cartoon and sticks and color the RMSD based on all the atoms, which would be a more accurate representation. Also, related to point #3 above it would be interesting to know if the sidechains that participate in inter-Fab contacts differ between the EM and x-ray structures.

---

## [Author Response]

Reviewer #1:[…] My only query is how does the binding change with the use of intact monoclonal antibodies? I do not need to see experimental data for this as I understand it would complicate the structural approaches. Nevertheless, either by vaccination or by antibody-based delivery, intact antibodies are binding. It would be good to hear about how this constrains the binding of repeats and if changes in the IgG backbone can be part of antibody design.1) For example, do you observe the same number of 3D11 molecules bound to PbCSP complex to be the same if Fab or intact antibody is used using SEC-MALS?2) Would you expect to see differences in cryo-EM if the intact mAb was used?

We thank the reviewer for her positive comments. We agree that the binding of intact monoclonal antibodies to CSP is an interesting and biologically relevant topic of research. However, binding of 3D11 IgG to PbCSP results in the formation of large aggregates due to antibody-mediated crosslinking. This process has previously been observed in the context of the parasite surface and is known as the circumsporozoite precipitation reaction (Yoshida et al., 1980; Potocnjak et al., 1980). Nevertheless, we attempted to biochemically characterize the 3D11 IgG-PbCSP complex using negative-stain electron microscopy and we have now included this data in (Figure 5—figure supplement 2). Based on the 2D analysis, we conclude that 3D11 IgG-PbCSP complexes have a similar architecture to 3D11 Fab-PbCSP complexes; however, the heterogeneity of the 3D11 IgG-PbCSP complexes precluded us from obtaining 3D reconstruction. To describe these results, we have added the following sentences: “To examine whether 3D11 IgG can induce a similar type of spiral conformation of PbCSP as 3D11 Fab, we prepared complexes of 3D11 IgG-PbCSP for negative-stain (ns) EM analysis. […] Our findings are in agreement with a similar analysis previously performed with human 311 Fab and IgG in complex with PfCSP, which also observed the ability of both IgG and Fab to induce a spiral-like conformation in CSP.”

Reviewer #3:[…]1) From the way it is presented, it is hard to fully appreciate the details of the cryo-EM structure of the 3d11-PbCSP complex, particularly with regards to the inter-Fab contacts. It would be helpful to describe the general locations of these residues within the Fab structure (CDRH/L 1,2,3; FR). This could be compared to the published 311-rsCSP structure in order to understand the structural properties of spiral-inducing antibodies in general. Also see comments on Figure 5 below.

We thank the reviewer for his suggestions. We have introduced additional panels and labels in Figure 5 to provide a more in-depth description of the residues engaging in inter-Fab contacts, and revised the subsection “Contacts between 3D11 Fabs stabilize the PbCSP spiral structure” to describe the positions of these residues of interest relative to the overall Fab structure. We also added the following sentences to the Discussion section: “In human mAbs 311 and 1210, CDR3 regions of both heavy and light chains appear to play a considerable role in forming Fab-Fab contacts. Interestingly, in the case of mAb 3D11, homotypic interactions are mainly mediated by residues localized in HCDR1 and -2, KCDR1, and FR3 regions of both the HC and KC, with little contribution from residues in the CDR3 regions (with the exception of H.Tyr97 in HCDR3 and K.Phe94 in KCDR3).”

2) From the binding studies, the authors conclude that 10 or 11 Fabs can bind one molecule of PbCSP. The 2D classes in Figure 4—figure supplement 2B also show what appear to be more than 7 Fabs in a complex. Did the authors use 3D classification to sort out these extended complexes, or are the "trailing" Fabs simply blurred out in the high resolution reconstruction? Did the authors see any reconstructions with a fully occupied CSP? In the Materials and methods, and/or in a supplementary figure, it would be useful to have a few more details of the data processing workflow in Cryosparc, which would inform of if and how the low affinity PPAP/NAND repeats can be occupied simultaneously with the PPPP/NPND repeats.

We have provided an additional supplementary figure (Figure 4—figure supplement 1) showing the cryoEM data processing workflow in cryoSPARC. Despite extensive 3D classification in cryoSPARC v2 we were unable to obtain 3D classes with varied number of Fabs bound. The final low-pass filtered (20 Å) cryoEM structure indicates ~15 Fabs bound to the PbCSP (Figure 4—figure supplement 2F), although the density of the N- and C-terminal Fabs is very weak. The results of 3D variability analysis in cryoSPARC suggest that the blurring of the distal Fabs is a result of the internal flexibility of the complex (see Figure 4—video 1).We have added the following sentences to describe these observations: “Although the low-pass filtered (20 Å) cryoEM map of the 3D11 Fab-PbCSP complex contains visible density for >10 3D11 Fabs (Figure 4—figure supplement 2F), only the density for the seven central Fabs was strong enough to warrant building a molecular model. Indeed, 3D Variability Analysin in cryoSPARC v2 revealed continuous flexibility at the N- and C-termini of the 3D11 Fab-PbCSP complex (Figure 4—video 1).”

3) Figure 5C. It is quite difficult to understand the structural details presented here. This figure would benefit greatly from making at least two panels which present two different views of the interface which focus on different details of the inter-fab contacts. Also, importantly, there is very little structural context outside of the side-chains shown, and the CDR loops to which these residues belong to are unlabeled. As mentioned above, it would be useful to know what general structural features of these antibodies can be utilized to drive high-avidity complex formation on CSP and by consequence stabilize long-range helices.

We thank the reviewer for his suggestions. We have introduced additional panels and labels to provide a more detailed representation of the complex.

4) Figure 4—figure supplement 3 is a little bit confusing. The authors state that the x-ray and cryoEM structures are remarkably similar, which it expected. However, as colored, there are some local changes that are 1.5-2.5 Å RMSD, which is actually fairly large. Presumably this is due to the calculation based on all atom, not backbone. I would expect the larger distances to be localized to side chains rather than backbone, which is colored. It may therefore be better to represent the structure as a cartoon and sticks and color the RMSD based on all the atoms, which would be a more accurate representation. Also, related to point #3 above it would be interesting to know if the sidechains that participate in inter-Fab contacts differ between the EM and x-ray structures.

We appreciate the concern raised by the reviewer. We have re-calculated the RMSD based on the backbone alone and revised the figure to show this ribbon representation. We have also included additional panels to highlight the all-atom RMSD of residues participating in inter-Fab contacts.